# Pilot-Scale Test Results of Electrodialysis Bipolar Membrane for Reverse-Osmosis Concentrate Recovery

**DOI:** 10.3390/membranes12010083

**Published:** 2022-01-13

**Authors:** Leyla Gazigil, Eren Er, O. Erdem Kestioğlu, Taner Yonar

**Affiliations:** 1Environmental Engineering Department, Faculty of Engineering, Gorukle Campus, Bursa Uludag University, 16059 Bursa, Turkey; leylagazigil@uludag.edu.tr (L.G.); erener90@gmail.com (E.E.); 2Uluçev Environmental Technologies Ind. Trade. Co., Ltd., Ulutek Technology Development Zone, Nilufer, 16285 Bursa, Turkey; ekestioglu@ulucev.com.tr

**Keywords:** electrodialysis bipolar membrane, concentrate, acid recovery, base recovery, membrane

## Abstract

In this study, it is aimed to investigate the potential of electrodialysis bipolar membrane (EDBM) systems for the recovery of the concentrate originating from an organized industrial estate (OIE) wastewater treatment system with reverse osmosis (RO). Acids and bases were obtained from a pilot-scale treatment plant as a result of the research. Furthermore, the sustainability and affordability of acids and bases obtained by EDBM systems were investigated. Six cycles were carried out in continuous-flow mode with the EDBM system as batch cycles in the disposal of the concentrate and the production of acids and bases with the EDBM system. For each cycle, the EDBM system was operated for 66, 48, 66, and 80 min, respectively, and the last two cycles were operated for a total of 165 min (70 + 90) with 5 min of waiting. In the EDBM system, a working method was determined such that the cycle flow rate was 180 L/hour, energy to be given to the system was 25 V, and the working pressure was in the range of 0.8–2.5 bar. In the six cycles with the EDBM system, the concentrate, acid and base, conductivity, pH, and pressure increase values were investigated depending on time. Throughout all these studies, the cycles were continued over the products formed in the acid and base chamber. As a result of all the cycles, acid (HCl) production at a level of 1.44% and base (NaOH) production at a level of 2% were obtained.

## 1. Introduction

Expansions of population and economic development have led to an increasing freshwater deficit [1]. Water and wastewater treatment efficiency, including recovery of resources, has become a top factor in the emergence of sustainable procedures for effective brine recovery and valorization [2]. This should result in the creation of new or better innovative water-treatment methods in real-world settings. Desalination technologies have received significant attention in recent years as a vital option to address water scarcity challenges all over the world [3]. For the time being, the widely known and used desalination technologies are reverse osmosis (RO), nanofiltration (NF), electrodialysis (ED), multistage flash distillation, and multiple-effect distillation. RO contributes about 65% of total installed desalination capacity worldwide [4]. The handling/treatment of the concentrated brine byproduct, on the other hand, is a key challenge in the use of RO desalination. The recovery rate of industrial RO desalination systems is normally about 50%. Therefore, over half of the feed water is released as saline into the sea or adjacent regions, with serious environmental consequences [5]. Membrane clogging is another issue with RO desalination. This problem occurs as a result of all strong acids (e.g., HCl or H_2_SO_4_) and bases (e.g., NaOH) utilized to adjust pH values or reagents used for cleaning [6]. Electrodialysis bipolar membrane (EDBM) is a technology that integrates bipolar membranes with traditional electrodialysis [7]. EDBM is a promising method for treating and valorizing desalination brines by producing acids and bases, which are valuable compounds in almost any desalination plant. EDBM has also been effectively utilized to produce or purify acids, as well as to adjust pH during fermentation or chemical synthesis in biochemical and food processing [8,9,10]. Furthermore, there are some applications in the removal of heavy metals and exhaust emissions [11,12,13]. 

Electrochemical processes such as ED and EDBM can contribute to soft-water production and the evaluation of waste fluxes [14]. EDBM is a new technology that combines the separation function of electrodialysis with water separation at the bipolar membrane interface, which can convert salts into corresponding acids and bases without adding external components [15]. In this system, anions and cations are separated from wastewater separately and combined with H^+^ and OH^−^ ions via bipolar membranes to form acidic and alkaline solutions [16].

Briefly, the advantages of the ED system are: a small and simple pre-treatment is sufficient before the process; low operating pressure; no need for antiscalant (membrane protector); long membrane life; low operating and maintenance costs; effective on many ion forms; efficacious in waters with high ion content (10,000 mg/L TDS); obtaining output product in the order of 90% of the input-product water; 10% concentrate formation (advantage over RO); it is approximately 5 times longer-lasting than reverse osmosis (reverse osmosis, 1–2 years; ED, 8–10 years); two separate collections of concentrated products and ease of recovery; high selectivity for charged compounds. In addition to these huge benefits of EDBM, it has some cons, such as: electricity consumption; the need for expertise, qualified and trained personnel; ineffectiveness on microorganisms and most anthropogenic organic pollutants [17].

The EDBM process has been widely used for many years in the recovery of process water [18,19]. Xu et al. (2022) used EDBM to produce acids and bases from brine-industry wastewater. The maximum desalination, acid, and alkali production rates obtained in this study were 0.304 mol/h, 0.114 mol/h, and 0.136 mol/h, respectively [20]. Yuzer et al. (2021) used EDBM for wastewater and salt recovery in biologically treated textile wastewater. They found that the acid, base, and wastewater produced by the EDBM process can be reused in wet textile processes. [21]. Jiang et al. (2021) tested a combination of RO, ED, and EDBM for salt recovery and acid/base production for the treatment of cold-rolling wastewater. As a result of EDBM application, ED concentrated solution containing Na_2_SO_4_ was successfully desalinated for acid/base production and pure water production [22]. Rózsenberszki et al. (2021) used the bipolar membrane electrodialysis technique for itaconic acid (IA) recovery from fermentation wastewater. Experimental results indicated that the IA recovery rate/current efficiency was 74%/77% and 63/41% under the initial pH of 3.0 and 7.4, respectively [23]. Gössi et al. (2020) focused on the in situ recovery of carboxylic acids from fermentation liquors via membrane-assisted reactive extraction using membrane modules with enhanced stability [24]. Gao et al. (2021) used EDBM technology to treat waste sodium sulfate containing lithium carbonate to convert low-value sodium sulfate to high-value sulfuric acid and sodium hydroxide [25]. With EDBM, it is possible not only to recover acid-base but also to recover many different compounds with economic value. For example, Saabas et al. (2021) aimed to recover ammonia from simulated membrane-contactor wastewater using EDBM. They recovered up to 68% of ammonia [26]. Kuldeep et al. (2021) used EDBM for sulfate recycling in metallurgical industries [27].

Sustainable production of pure chemicals is one of the main goals of EDBM today. Based on this idea, Virag et al. demonstrated the development and optimization of a continuous and simultaneous isolation process for three biophenols based on temperature-shift adsorption. They concentrated the product and waste streams, recycled the solvent in-line, and investigated their impact on the E factor, carbon footprint, and economic sustainability of the process. As a result, they stated that the application of the hybrid process consisting of printing technology and nanofiltration can be extended from complex mixtures to the isolation of other natural compounds [28]. Carla et al., in their study, aimed to isolate the mixture of enantiomers by means of a membrane electrodialysis cell as the final step of a new technology to obtain dex-ibuprofen-enriched ibuprofen. They developed an electrodialysis cell consisting of four compartments separated by cation, anion, and cation exchange membranes, allowing selective migration of ibuprofen by preventing its dissociation on the electrodes. In fact, stainless-steel electrodes immersed in ammonium-formate solutions were separated by cation exchange membranes from chambers containing prophene (~200 ppm in ethanol, pH 6.81–6.34). The best separation performance was achieved after 6 h of operation at 60 V with an anion exchange membrane that resulted in up to 57% extraction of ibuprofen from the products of enzymatic esterification in ethanol medium. The membrane allowed selective migration of both stereoisomers and negligible amounts of ethyl esters. They argued that the presented approach of this research is innovative as the electrodialysis process allows the design of an environmentally friendly technology to be completed [29]. Levente et al. reported the characterization and an easy fabrication method of nanocomposite anion exchange membranes (AEM) based on modified graphene oxide (GO) and quaternized polybenzimidazole (PBI). PBI was chosen for its outstanding thermal and mechanical stability and good film-forming properties. They expected GO-polybenzimidazolium nanocomposite AEMs prepared in their study to have improved mechanical and electrochemical properties. GO-polymer interactions, the effects of filler loading (0.25–2.5%), GO distribution, and membrane morphology were systematically investigated to reveal structure-property relationships. They also proposed a Robeson-type plot for AEMs to compare commercial and published AEMs [30].

In this study, acid and base production studies were carried out from the concentrate originating from the RO system by using an electrodialysis bipolar membrane (EDBM). It was aimed to produce acids and bases with an economic value by disposing the concentrate formed from the reverse-osmosis system, which is harmful to the environment. Thus, the concentrate, which should be disposed of as hazardous waste, will be recycled in order to approach zero waste. Thus, there will be a great economic gain for industries. At the same time, by using the EDBM system for concentrate disposal, natural resources will be preserved, and an environmentally friendly approach will be displayed. The feasibility, sustainability, and economy of the concentrate, which is first disposed of with an innovative technology, such as EDBM, and then the re-use of the acids and bases obtained as a product in treatment plants or by transforming them into products used in the industry will be demonstrated.

## 2. Materials and Methods

### 2.1. Wastewater Supply and Characteristics

The wastewater used in the study was supplied from the organized industrial estate wastewater treatment plant (OIE WWTP) based in Bursa, Turkey, which treats 60,000 m^3^/day of domestic and 40,000 m^3^/day of industrial wastewater. A total of 34,000 m^3^/day of industrial wastewater originates from the textile industry, while the remaining 6000 m^3^/day arises from other industries.

In the study, the wastewater given to the Advanced Pilot Wastewater Treatment Plant (APWWTP) was obtained from the chlorine-contact tank outlet of the OIE WWTP. The wastewater was read 3 times, and the average of the analysis results of the wastewater taken from the outlet point is given in Table 1.

### 2.2. Pilot WWTP Concentrate Production and Disposal System Equipment 

Field studies were carried out at the APWWTP, with a capacity of 5 m^3^/h. APWWTP used in the study encompasses pressure sand filter (PSF), TF-716 microfiltration (MF), mini UF-4 ultrafiltration (UF), and TFZ 29000 reverse-osmosis (RO) systems. The concentrate resulting from the reverse-osmosis process is fed to the EDBM system for acid-base production. The flow chart of the pilot plant is illustrated in Figure 1.

#### 2.2.1. Pilot WWTP Concentrate Disposal System Equipment

Afterwards, the investigation of acid-base production with the obtained concentrate was carried out in a pilot facility that enfolds FILMTEC NF-2540 nanofiltration (NF) and electrodialysis bipolar membrane (EDBM) systems.

#### 2.2.2. Nanofiltration (NF) 

Nanofiltration has a pore diameter of approximately 0.001 μm. The molecular separation limit of the membranes is between 180 and 2000 Da. Thus, components with this molecular weight can be easily isolated from components with higher molecular weights. The application pressure range is mostly 3–40 Bar. Many multivalent ions (such as sulfate-carbonate, M^+2^) are retained in the membrane, owing to the molecular separation between reverse osmosis and ultrafiltration, while diluted solutions of monovalent ions usually pass through the membrane unhindered [31].

#### 2.2.3. Electrodialysis Bipolar Membrane System (EDBM)

For the pilot study, the EDBM FT-ED 100 module, the membrane and spacers applied in electrodialysis, were obtained from FUMA-TECH GmbH (Germany). Glass, tube, and float-flow meters measuring the flow, as well as SEKO brand mechanical diaphragm dosing pumps providing the flow, were obtained from a private company. In addition, MCH 34050 model direct-current power supply is used to supply current to the cell. The technical specifications of the EDBM system are given in Table 2. The working cycle of the EDBM system is illustrated in Figure 2.

#### 2.2.4. EDBM System Performance Calculation

The formula applied while calculating the performance of the EDBM system is provided in Equation (1) [33,34].
(1)$$I_{e}=\frac{96500\times \left(\;V_{i}\times C_{i}-V_{f}\times C_{f}\;\right)}{I_{d}\times S\times t}$$

*I_e_*: Available current efficiency)*V_i_*: Inlet volume of acid or base (Liters)*C_i_*: Concentration of input acid and base (equivalents/ L)*V_f_*: Final acid and base volume (Liters)*C_f_*: Final acid and base concentration (equivalents/ L)*I_d_*: Current density (A/m^2^)*t*: Times (s)*S*: Active membrane area (m^2^).

### 2.3. Measurement and Analysis Methods 

Some anions and cations in the concentrate prevent the EDBM system from working efficiently. The results of the concentrate analysis and the scientific studies revealed that the concentrate has M^+2^ valence ions, and these ions (especially Ca^+2^, Mg^+2^, and Ba^+2^) were determined to cause pressure rise and blockages in the EDBM system [35,36]. In addition, it is known that anions (NO_3_^−^, SO_4_^−2^, and Cl^−^) in the concentrate, which are effective in acid production, cause the formation of different acids.

In order to remove the M^+2^ valent metals contained in the obtained concentrate, while the monovalent anions (NaCl, CaCl_2_) are removed by 20–80%, the nanofiltration (NF) system, which removes the divalent anions (such as MgSO_4_) by 90–98%, has been added to the pilot plant. Within the scope of this study, analyses were made to determine the inlet wastewater characteristics, the efficiency of the plant by running the pilot plant, the efficiency of the NF system, and the M^+2^ removal efficiency. Na^+2^, Ca^+2^, Mg^+2^, NO_3_^−^, SO_4_^−2^, Si^+2^, Cl^−^, Fe^+2^, and Ba^+2^ analyses are made in accredited laboratories within the scope of relevant standards in order to reveal the M^+2^ removal efficiency of the NF system [35,36]. The analyses and related standards made within the scope of all studies are given in Table 3.

The conductivity value of the RO sourced concentrate was read in the range of about 4000 μS/cm. This value is not sufficient to enter the EDBM system. By adding synthetic NaCl to the RO concentrate, the conductivity value was ameliorated from 4000 μS/cm to 8000 μS/cm, and 200 L NF filtrate was obtained by passing it through the NF system. 

Samples of the concentrate originating from RO, the concentrate with NaCl added (given to the NF system), the NF outlet concentrate originating from the NF system, and the samples belonging to the NF outlet filtrate were analyzed in accredited laboratories, and the analysis results are given in Table 3. Analyses were performed according to the standart methods specified in Table 3.

Conductivity and pH parameters are the main variable parameters to reveal the performance of the system [37,38]. Therefore, in the study, instantaneous conductivity, pH, and pressure changes were measured at certain intervals after the EDBM system was started. pH and conductivity measurements were made with the Hach HQ40D protractive pH and conductivity meter. In the vicinity of the main reading module of the device, there are CDC401033 conductivity probes with cables and a liquid-filled pH probe with a PHC301033 cable. The monitoring of pressure changes was carried out with pressure gauges on the system.

In the pilot plant studies, approximately 1 ton of water treated with conventional treatment methods was taken from the WWTP chlorination unit outlet by means of a submersible pump and given to the pilot WWTP. After the pilot plant was run, samples were taken from the inputs and outputs of the PSF, MF, UF, and RO systems in order to demonstrate the efficiency of all equipment in the system, and pH, temperature, conductivity, SS, and COD analyses were carried out.

After the concentrate production and demonstrating system efficiency in the pilot plant, acid and base production was started from the concentrate with the NF and EDBM system.

### 2.4. Determination of EDBM System Operating Conditions 

The EDBM system was operated in an intermittent cycle for the disposal of the concentrate and the production of acids and bases. In the EDBM system, a working method was determined such that the cycle flow rate was 180 L/hour, the energy to be given to the system was 25 V, and the working pressure was in the range of 0.8–2.5 bar. In addition, in the studies conducted with the EDBM system, it was attempted to keep the average conductivity of the feed water in the range of 6000–7400 μS/cm at the beginning of the cycles. The experiments were carried out at ambient temperature.

In the study, 6 cycles were performed in continuous flow mode in a three-chamber electrodialysis cell with a bipolar membrane. The EDBM system for each cycle was 66, 48, 66, 80, 70, and 90 min, respectively.

In the experiments, conductivity, pH, pressure, and current values were considered as variables. The conductivity and pH values of acids, bases, salts, and Na_2_SO_4_ formed during the pilot plant studies with the whole EDBM system were measured instantaneously at certain time intervals (between 5 and 10 min). Pressure changes were instantly measured and recorded with the pressure gauge on the system. In addition, current changes from the top of the direct current source were read and recorded at the same time intervals.

## 3. Results and Discussion

### Concentrate Disposal Studies and Analysis Results with EDBM System 

A total of six cycles were made with the intermittent EDBM system from RO-sourced concentrate. The main target is to convert the ions in the concentrate into acid-base ions via cationic, anionic, and bipolar membranes. In the first cycle, the acid-base-forming ions in the concentrate transferred to acid-base units, and when the conductivity value in the concentrate reached 732 μS/cm, the concentrate was discharged, but the acid-base units continued to cycle with the same volume. At the end of each run, concentrate, acid and base, Time-pH change, time-conductivity change, time-pressure change, and time-conductivity-ampere change graphs were obtained. The data of the study results are given in Figure 3, Figure 4, Figure 5, Figure 6, Figure 7 and Figure 8 from the first to the sixth trial. In the experiments, the energy given to the system is 25 V, and the cycle flow rate is 180 L/s.

In the first trial, acid-base production was made from the concentrate passed through the NF system with the EDBM system for 66 min, and the concentrate, acid and base, conductivity, pH, and pressure increase values were investigated depending on time. When the results obtained in the studies are examined, while the conductivity values in the concentrate decrease, the produced acid—base conductivity values increase. Since Na_2_SO_4_ is used to protect the electrodes, no change in conductivity values is observed. However, the increase in conductivity in base production is lower than in acid production. This is thought to be due to the M^+2^ valence ions still present in the concentrate. It has been demonstrated by other studies that M^+2^ valence ions can accumulate in the system and cause an increase in resistance on the membrane surface, causing concentration polarization and clogging [39]. Blockages in the system both reduce product quality and increase operating time. In the preliminary study with the concentrate, it was determined that the concentrate causes pressure rise and blockages in the base tank due to the M^+2^ valence ions it contains. When the time-dependent pressure changes are examined, it is observed that there is no big difference for this experiment. When the time-dependent pH change graph is examined, it is observed that it is at pH 12 levels in base production and at 0 levels in acid production. In the EDBM system, the first 30 min. of the amperage change against time. A decrease is observed at the end of the 30th minute. In EDBM systems, the voltage supplied to the system is constant. When the conductivity value of the concentrate decreases, since the voltage supplied to the system is constant and the resistance changes with the conductivity, the current passing through the system also changes (V = I × R) [19].

In the second trial, acid-base production was made from the concentrate passed through the NF system with the EDBM system for 48 min, and the concentrate, acid and base, conductivity, pH, and pressure increase values were investigated depending on time. When the results obtained in the studies are investigated, as in the first trial study, the conductivity values in the concentrate decrease, while the produced acid-base conductivity values increase. Since Na_2_SO_4_ is used to protect the electrodes, no change in conductivity values is observed. However, the conductivity increase in base production is lower than in acid production. This is esimated to be due to the M^+2^ valence ions still present in the concentrate. The effect of ions has been demonstrated by previous studies [40]. When the time-dependent pressure changes are examined, it is observed that there is no big difference for this experiment. When the time-dependent pH change graph is investigated, it is observed that it is stable at pH 12 levels in base production and at 0 levels in acid production. In the EDBM system, during the first 15 min of the amperage change against time, it is observed that a decrease starts at the end of the 15th minute.

In the third trial, acid-base production studies were carried out with EDBM systems from the concentrate passed through the RO and NF systems, and acid-base production was carried out for 66 min. The concentrate, acid and base, conductivity, pH, and pressure increase values were investigated depending on time. When the results obtained in the studies are examined, as in the first and second trial studies, the conductivity values in the concentrate decrease, while the produced acid-base conductivity values increase. Since Na_2_SO_4_ is used to protect the electrodes, no change in conductivity values is observed. However, the conductivity increase in base production is lower than in acid production. This is appraised to be due to the M^+2^ valence ions still present in the concentrate. When the time-dependent pressure changes are investigated, it is observed that there is no big difference for this experiment. When the time-dependent pH change graph is examined, it is observed that it is stable at pH 12 levels in base production and at 0 levels in acid production. In the EDBM system, it is observed that the amperage change against time increases in the first 20 min, and a decrease begins at the end of the 20th minute.

In the fourth trial, acid-base production was made from the concentrate passed through the NF system with the EDBM system for 80 min, and the concentrate, acid and base, conductivity, pH, and pressure increase values were investigated depending on time. In this experiment, work was continued with the concentrate with a conductivity of 7200 µS/cm on the acid-base obtained in the third trial. When the time-dependent pressure changes are investigated, it is observed that there is no big difference until the 70th minute for this trial. When the time-dependent pH change graph is examined, it is observed that it is stable at pH 12 levels in base production and at 0 levels in acid production. In the EDBM system, during the first 10 min. of the amperage change against time, it is observed that a decrease starts at the end of the 10th minute.

In the fifth trial, the studies were continued with the concentrate passed through the NF system. The EDBM system was operated for 70 min. When the concentrate conductivity reached 747 µS/cm, acid conductivity reached 79,400 µS/cm, and base conductivity reached 47,800 µS/cm, the concentrate was discharged, and the sixth trial was started.

In the sixth trial, the concentrate conductivity was again entered as 7400 µS/cm, and the EDBM system was operated for 90 min. When the concentrate conductivity reached 850 µS/cm, the system was stopped. Stopping the system at the 90th minute was due to the decrease in conductivity at certain times. Since the acid conductivity is around 89,700 µS/cm and the base conductivity is around 60,800 µS/cm, the acid-base formation time from the concentrate has increased, and an increase in pressure was observed in the base formation unit in the EDBM system.

The acids and bases produced as a result of all cycles were determined using the calculation method in Equation (1). As a result of all cycles, acid (HCl) production of 1.44% and base (NaOH) production of 2% were realized. Wisniewski et al. have obtained similar results in their studies [36,41]. In light of these data, the study was terminated after the sixth trial due to the pressure increase in the system at the end of the sixth trial and the decrease in conductivity in the acid tank.

## 4. Conclusions

In the studies, it was observed that the maximum acid-base conductivity (90,000 µS/cm in acid, 60,000 µS/cm in base) was reached with the EDBM system using only the salt in the concentrate. Relating concentrations correspond to approximately 1.44% acids and 2% bases. Quantities were obtained after a 4-h study and continuous circulation of acid-base. At this point, it has been determined that the percentages of acids and bases formed cannot be increased furthermore under the current conditions (with the existing pilot plant with a membrane surface area of 100 cm^2^, a maximum voltage of 25 volts, and a maximum current of 3 amps).

Even if the acid and base production concentration is low, many treatment plants (including the existing plant) use acids and bases. The acids and bases produced here have the potential to contribute to the system’s own needs. In other words, even if the concentration is low, there may be a potential for use, at least in the existing facility or in nearby businesses. Thus, the opinion that EDBM is one of the best sustainable alternatives in the management of reverse-osmosis concentrate, which is a critical problem for the environment, has been demonstrated by this study.

As a result, it has been determined by the studies that wastewater can be more economically recovered. Concentrate originating from reverse osmosis is a method that can be used for acid and base production, but measures such as salt addition should be taken in order to ensure its economic use. However, it is thought that it will be possible to produce acids and bases at desired concentrations with only reverse-osmosis concentrate, without adding salt, by modifying the study, together with rapidly developing membrane technologies.

In line with the results of this study, it is aimed to increase the current efficiency with further studies. With this study, a database will be created for researchers who work in this field since, as a result of the research conducted within our scope regarding the production of acids and bases from the concentrate formed after RO, we have not encountered a similar study.

This study was carried out for the sustainable disposal of the concentrate formed as a result of the reverse-osmosis process. Acids and bases were produced from this concentrate using EDBM systems. It is thought that higher acid-base amounts can be achieved, especially from wastewater with higher salinity. The approach presented in this research has shown that it is possible to dispose of the concentrate formed as a result of reverse osmosis with an environmentally friendly technology. Thus, it is thought that it will be a guide for future studies.

## Figures and Tables

**Figure 1 membranes-12-00083-f001:**
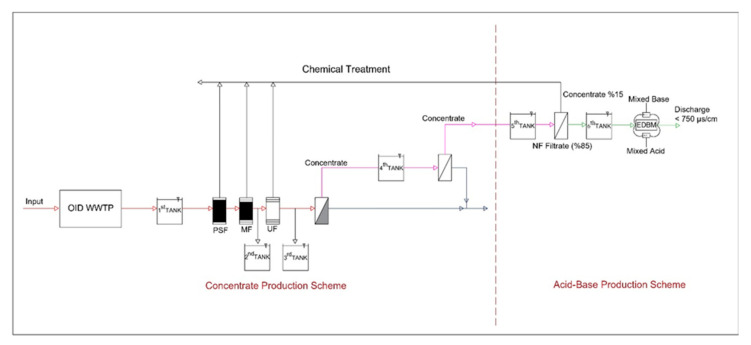
Flow chart of the pilot plant where the concentrate from the RO system and the acid-base are obtained.

**Figure 2 membranes-12-00083-f002:**
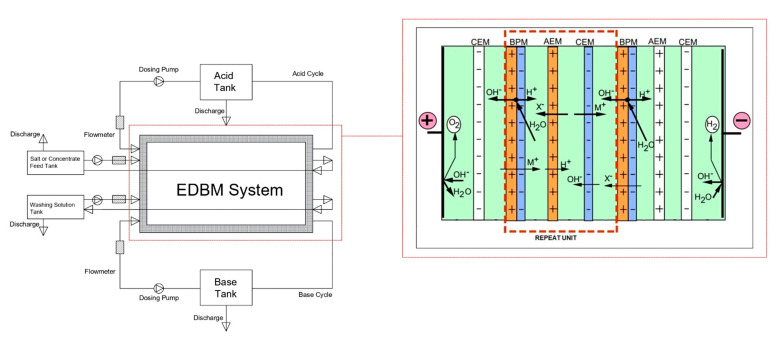
The operating cycle of the EDBM system [32].

**Figure 3 membranes-12-00083-f003:**
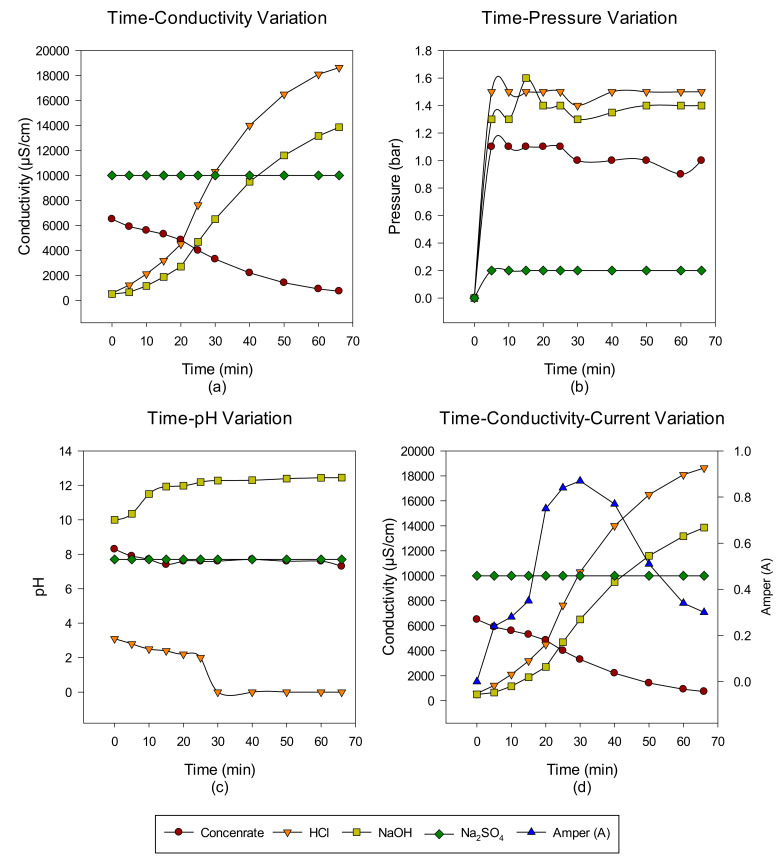
NaCl–NaOH–HCl–Na_2_SO_4_ (**a**) conductivity, (**b**) pressure, (**c**) pH, and (**d**) conductivity and ampere change between 0 and 66 min.

**Figure 4 membranes-12-00083-f004:**
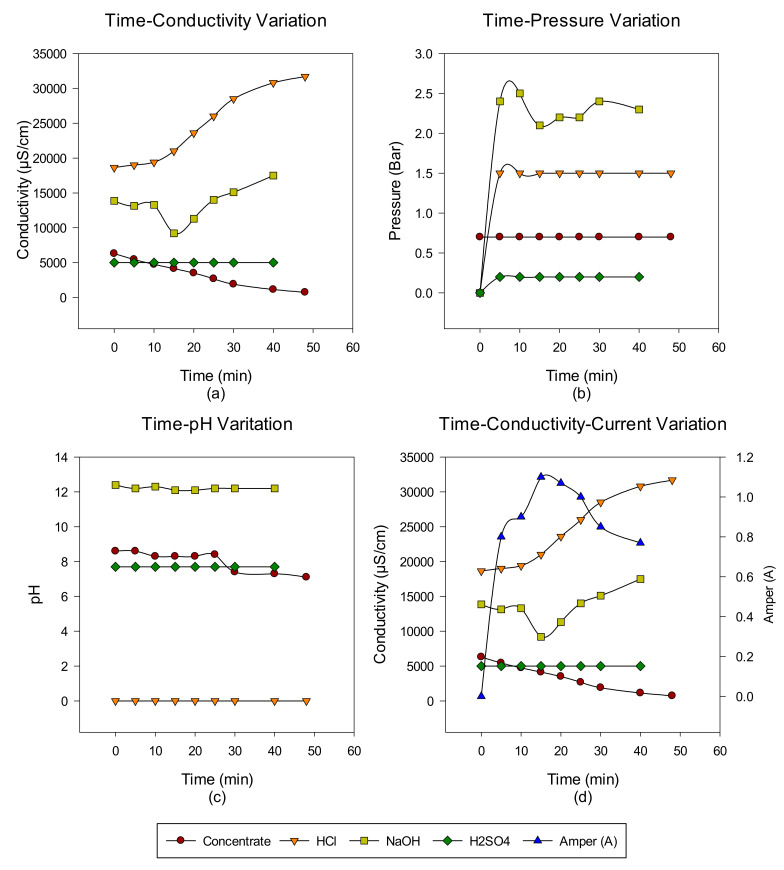
Between 0 and 48 min, NaCl–NaOH–HCl–Na_2_SO_4_ (**a**) conductivity, (**b**) pressure, (**c**) pH, and (**d**) conductivity and ampere change.

**Figure 5 membranes-12-00083-f005:**
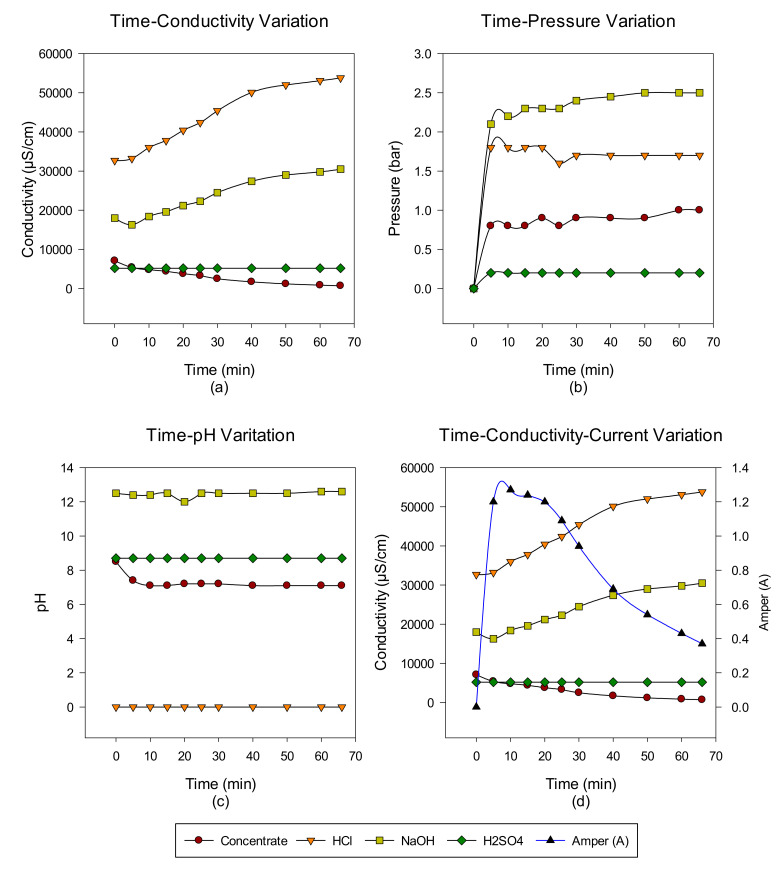
NaCl–NaOH–HCl–Na_2_SO_4_ (**a**) conductivity, (**b**) pressure, (**c**) pH, and (**d**) conductivity and ampere change between 0 and 66 min.

**Figure 6 membranes-12-00083-f006:**
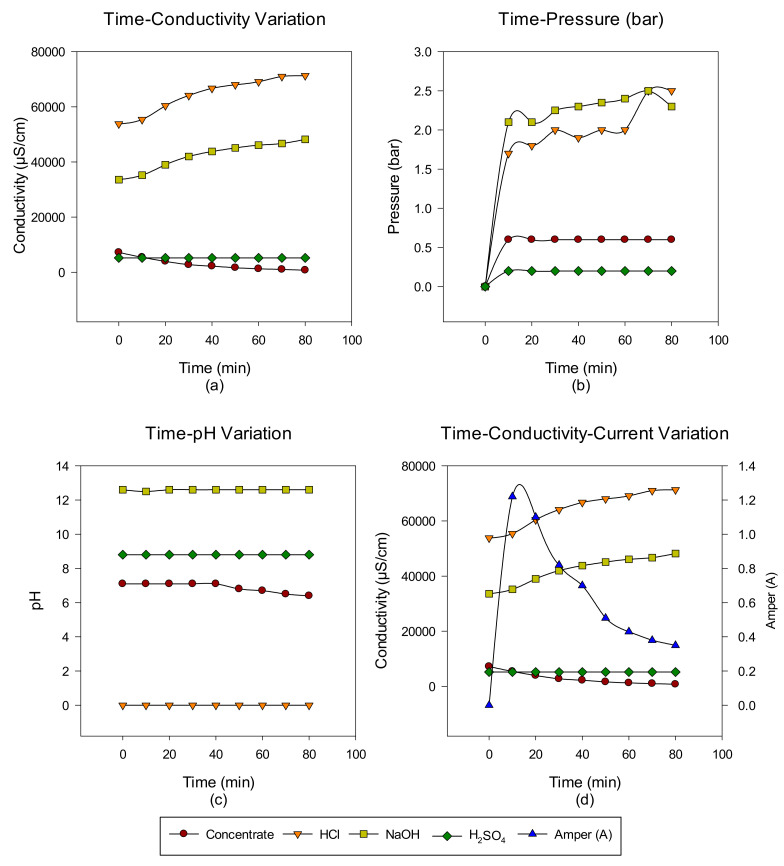
Between 0 and 80 min, NaCl–NaOH–HCl–Na_2_SO_4_ (**a**) conductivity, (**b**) pressure, (**c**) pH, and (**d**) conductivity and ampere change.

**Figure 7 membranes-12-00083-f007:**
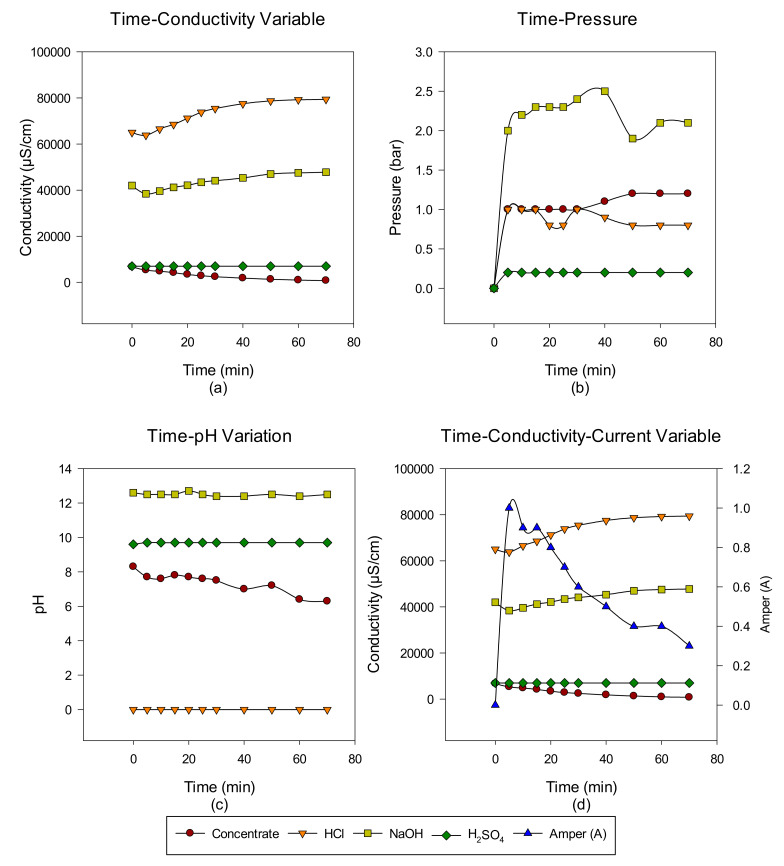
NaCl–NaOH–HCl–Na_2_SO_4_ (**a**) conductivity, (**b**) pressure, (**c**) pH, and (**d**) conductivity and ampere change between 0 and 70 min.

**Figure 8 membranes-12-00083-f008:**
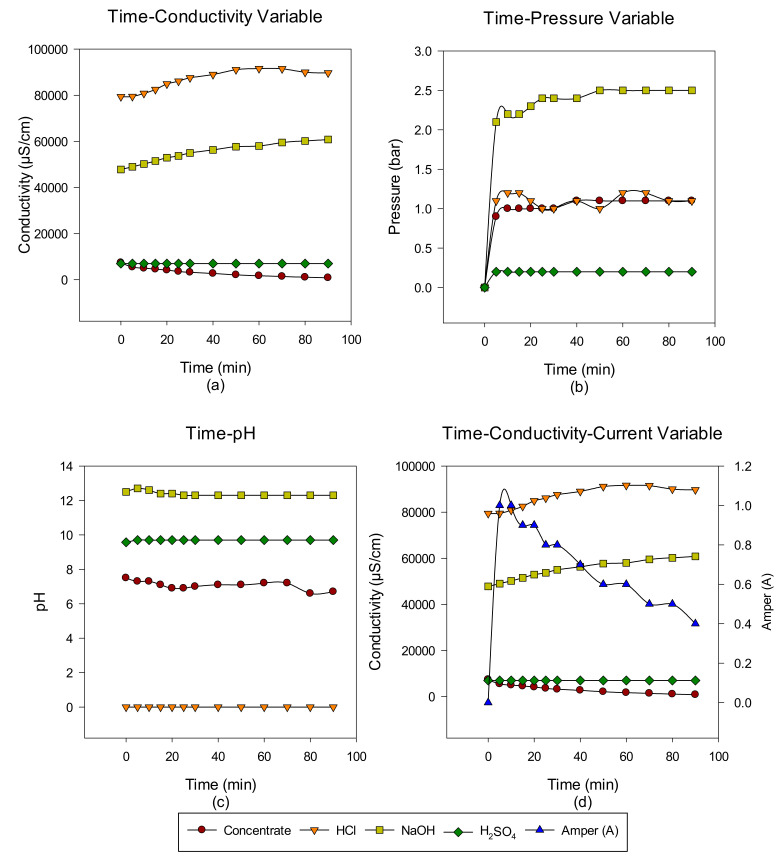
Between 0 and 90 min, NaCl–NaOH–HCl–Na_2_SO_4_ (**a**) conductivity, (**b**) pressure, (**c**) pH, and (**d**) conductivity and ampere change.

**Table 1 membranes-12-00083-t001:** Output values of OIE WWTP and pilot-plant processes.

Parameter	OIE WWTP Outlet	PSF Outlet	MF Outlet	UF Outlet	RO Outlet
pH	6.3 ± 0.15	8.06 ± 0.23	8.18 ± 0.21	8.24 ± 0.26	8.5 ± 0.12
Temperature (°C)	22.5 ± 0.53	21.4 ± 0.56	24.7 ± 0.85	21.6 ± 0.53	22.4 ± 0.18
Conductivity (µS/cm)	3906.0 ± 148	2750.0 ± 55.0	2760.0 ± 62.0	2760.0 ± 41.0	4510.0 ± 212.0
SS (mg/L)	48.7 ± 2.93	16.0 ± 0.92	6.0 ± 0.18	4.0 ± 0.13	4.0 ± 0.11
COD (mg/L)	105.1 ± 2.32	85.0 ± 1.54	100.0 ± 2.05	70.0 ± 1.85	106.0 ± 2.20

**Table 2 membranes-12-00083-t002:** Technical characteristics and specifications of the EDBM system.

Equipment Type	Unit	Number	Specifications
System Capacity	L/h/cell	10	180 L/h
			(10 cells, 18 L/h)
System Pressure	bar	-	0.8 bar
Membranes			
Membrane Size	mm	10	160 × 160
Effective Membrane Area	cm^2^	1	100
Membrane Cover	-	20	PVC/PET
		10	PE/PVDF
Electrode	-	1	Anode
	-	1	Cathode

**Table 3 membranes-12-00083-t003:** Wastewater analyses and methods made within the scope of the study.

Parameter	Unit	Method	RO Outlet	Reinforced with NaCl	NF System Output	NF System Output
			(Concentrate)	(8000 μS/cm)	(Filtrate)	(Concentrate)
pH		SM 4500 H+B	8.85	-	8.22	8.34
Conductivity	μS/cm	SM 2510.B	4100	8000	7300	20,940
Na^+2^	mg/L	* ISO 11885 ICP	686	1463	1211	3271.5
Ca^+2^	mg/L	* ISO 11885 ICP	60	49.95	20,4	189
Mg^+2^	mg/L	* ISO 11885 ICP	38.7	44.8	20.6	212
NO_3_^−^	mg/L	* TS 6231	38.5	36.5	37.4	43.7
SO_4_^−2^	mg/L	* TS 5095	593	703.25	31.7	3699
Si^+2^	mg/L	S.M 4500 SiO_2_.C	22.7	16.125	12	48.05
Cl^−^	mg/L	* SM 4500 Cl^−^ B	689	2972.4	2927.5	3231.5
Fe^+2^	mg/L	* ISO 11885 ICP	-	0.84	0.04	4.5
Ba^+2^	mg/L	* ISO 11885 ICP	-	˂0.1	˂0.1	˂0.1
Colour	Pt-Co	SM 2120C	-	-	16.9	-

* Analyses made in accredited laboratories.

## Data Availability

Not applicable.

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
