# Peer review of "Pilot-Scale Test Results of Electrodialysis Bipolar Membrane for Reverse-Osmosis Concentrate Recovery"

_membranes, 2022, doi:10.3390/membranes12010083_

Round 1
Reviewer 1 Report
(1) It is necessary to list the composition of the acid chamber and alkali chamber in the bipolar membrane system at the beginning of operation. At the same time, the authors need to explain the reason for configuring the acid chamber solution and the alkali chamber solution in this way.
(2) “In EDBM systems, the voltage supplied to the system is constant. When the conductivity value of the concentrate decreases, since the voltage supplied to the system is constant and the resistance changes with the conductivity, the current passing through the system also changes (V=I*R).” The authors attributed the change in current to the resistance. The authors need to provide data on resistance changes during system operation to support this explanation. When analyzing the resistance, the authors need to comprehensively consider the changes in the water inlet, acid chamber and alkali chamber solution. At the same time, the authors need to explain in detail why the current rose first and then fell.
(3) In the second trial, why did the conductivity of the alkali chamber first drop and then rise?
(4) Compared with the second trial, the current drop point of the third trial was advanced. What is the reason?
(5) The entire system had run for a total of six trials. When analyzing, the authors must focus on highlighting the phenomenon that is different from the previous trial, and at the same time explain in detail the reason for the phenomenon, rather than repeating the same data.
(6) It is recommended that the authors provide the material balance calculation diagram of the entire process, including flow rate, conductivity, main product types and concentrations, etc.
(7) The authors need to conduct an economical analysis of the entire system, including operating costs, generating acid-base benefits, and so on. At the same time, a comparison should be made with the existing reverse osmosis concentrated brine treatment process to highlight the characteristics and advantages of the process.
Reviewer 2 Report
- The introduction is very short and limited. The authors should elaborate on all aspects of the research and recent developments.
- The reproducibility of the work should be demonstrated by independent experiments and error bars added on the figures presented under the results section of the manuscript.
- What are the commas in Table 1 and in other numbers? It seems that they should be dots for decimal places? Clarify and correct accordingly. Also, appropriate units and symbols should be used, e.g. ‘oC’ is not the scientifically correct way to write the unit of the temperature.
- When errors are provided, the authors should clarify in the table/figure caption how the errors were derived. Are those standard deviations? Based on how many samples? How were the samples collected, i.e. independent experiments/samples?
- Continuous membrane processes aiming at recovery, should be better introduced, and diverse examples given as it is an important filed (10.1016/j.seppur.2020.116694; 10.1021/acssuschemeng.9b04245).
- The transferability of the results should be discussed. Is this an isolated case study or there are general conclusions and results that are of interest and use to other researchers? If yes, from which field, and what are those results presented in the manuscript? Add some discussions on the potential impact of the work.
- Figures 2 and 3 should be more informative by incorporating process parameters and ranges.
- Membrane electrodialysis examples from various fields should be given to introduce the topic (10.1016/j.memsci.2021.120081; 10.1039/C8TA09160A; 10.1016/j.memsci.2020.118714).
- Both the quotient (“x/y”) and negative exponent (“x y-1”) formats are used in the manuscript for units. Either of them should be used consistently, preferably the negative exponent format, which is recommended by the IUPAC. In some cases superscripts are also ignored.
- The limit of detection and limit of quantification should be reported for all the analytical techniques and methods that were used to quantify the concentrations in the collected samples.
Reviewer 3 Report
Manuscript entitled “Pilot Scale Test Results of Electrodialysis Bipolar Membrane for Reverse Osmosis Concentrate Recovery” submitted by Leyla Gazigil, Eren Er, Erdem KestioÄŸlu and Taner Yonar, can be considered for publication in Membranes Journal, after a major revision.
Here is a list of my specific comments:
- General comment: The utility of this study should be clearly highlighted in the manuscript.
- Page 1, Keywords: Replace “EDBM” with “electrodialysis bipolar membrane” here.
- Page 1, 1. Introduction: This section is too brief and should be detailed in order to describe the state of art in this field. Also, at the end of Introduction, the main objectives of this study should be clearly and detailed presented.
- Page 1, line 28: “Water and wastewater treatment efficiency…”. Add here as reference the paper Sequential Treatment of Paper and Pulp Industrial Wastewater: Prediction of Water Quality Parameters by Mamdani Fuzzy Logic Model and Phytotoxicity Assessment, Chemosphere, 227, (2019), 256-268, because it is relevant for this observation.
- Page 2, line 65: “Organized Industrial Estate Wastewater Treatment Plant (OIE WWTP)”. The geographical localization of this source of wastewater should be mentioned here.
- Page 2, line 81: Replace “plant is illustrated in Figure 2” with “plant is illustrated in Figure 1”.
- Page 4, Figure 4: This figure should be moved into Supplementary materials.
- Page 5, 2.3. Measurement and Analysis Methods: Provide in this section a properly description of the analytical methods used in this study.
- Page 7, 3. Results: Replace this title by “Results and discussion”. Also, all the experimental results included in this section must be more detailed discussed, in accordance with the main objectives of this study.
- Page 7, line 204: Replace “in Figure 5, Figure 6, Figure 7, Figure 8, Figure 9 and Figure 10 “ with “Figures 5-10”.
- Page 14, References: The number of references is quite low and must be increased.
Reviewer 4 Report
The paper “Pilot Scale Test Results of Electrodialysis Bipolar Membrane 2 for Reverse Osmosis Concentrate Recovery”
authors: Leyla Gazigil, Eren Er, Erdem KestioÄŸlu and Taner Yonar, presents an interesting topic to Membranes readers, but further corrections are necessary.
My principal questions or remarks:
The title is clear. The content is in accord with title.
The size of the article is appropriate to the contents.
The authors must underline the major findings of their work and explain how the use of their proposed procedures represents a progress to other similar studies. The novelty must be pointed.
The Abstract can be revised.
The key words permit found article in the current registers or indexes. Please don’t use abbreviations in keywords.
In the introduction isn’t clearly described the state of the art of the investigated problem. More references from the last years must be cited. It is this study actual?
The methods are well described and the equipment and materials have been adequately described.
The tables contain necessary results.
Please provide comparison with other studies.
The figures have a good quality.
The Conclusion must been better justified.
Please provide minimum 2 references from this journal (last years), for demonstrated that manuscript is in Membranes topic.
The literature isn’t sufficiently critical, current, and internationally evaluated.
The paper has the text presented and arranged clearly and concisely.
The paper was written in standard, grammatically correct English, small corrections are necessary.
Please complete citation on the first page.
Please respect author guide.
Author Contributions: For research articles with several authors, a short paragraph specifying their individual contributions must be provided. The following statements should be used “Conceptualization, X.X. and Y.Y.; methodology, X.X.; software, X.X.; validation, X.X., Y.Y. and Z.Z.; formal analysis, X.X.; investigation, X.X.; resources, X.X.; data curation, X.X.; writing—original draft preparation, X.X.; writing—review and editing, X.X.; visualization, X.X.; supervision, X.X.; project administration, X.X.; funding acquisition, Y.Y. All authors have read and agreed to the published version of the manuscript.” Please turn to the CRediT taxonomy for the term explanation. Authorship must be limited to those who have contributed substantially to the work reported.
Funding: Please add: “This research received no external funding” or “This research was funded by NAME OF FUNDER, grant number XXX” and “The APC was funded by XXX”. Check carefully that the details given are accurate and use the standard spelling of funding agency names at https://search.crossref.org/funding. Any errors may affect your future funding.
Data Availability Statement: In this section, please provide details regarding where data supporting reported results can be found, including links to publicly archived datasets analyzed or generated during the study. Please refer to suggested Data Availability Statements in section “MDPI Research Data Policies” at https://www.mdpi.com/ethics. You might choose to exclude this statement if the study did not report any data.
Acknowledgments: ……
Conflicts of Interest: Declare conflicts of interest or state “The authors declare no conflict of interest.” Authors must identify and declare any personal circumstances or interest that may be perceived as inappropriately influencing the representation or interpretation of reported research results. Any role of the funders in the design of the study; in the collection, analyses or interpretation of data; in the writing of the manuscript, or in the decision to publish the results must be declared in this section. If there is no role, please state “The funders had no role in the design of the study; in the collection, analyses, or interpretation of data; in the writing of the manuscript, or in the decision to publish the results”.
Please verify all references and respect authors guide. The journals must be abbreviated, there are words in other languages, for example sayı..
Round 2
Reviewer 1 Report
(1) It is necessary to list the composition of the acid chamber and alkali chamber in the bipolar membrane system at the beginning of operation. At the same time, the authors need to explain the reason for configuring the acid chamber solution and the alkali chamber solution in this way.
Figure 2 has been explained in more detail and the arrangement has been made.
(2) “In EDBM systems, the voltage supplied to the system is constant. When the conductivity value of the concentrate decreases, since the voltage supplied to the system is constant and the resistance changes with the conductivity, the current passing through the system also changes (V=I*R).” The authors attributed the change in current to the resistance. The authors need to provide data on resistance changes during system operation to support this explanation. When analyzing the resistance, the authors need to comprehensively consider the changes in the water inlet, acid chamber and alkali chamber solution. At the same time, the authors need to explain in detail why the current rose first and then fell.
The voltage supplied to the system in studies is constant (V=25) available in the manuscript. The current changes of each cycle are indicated in the graphs d) of the figures.
(3) In the second trial, why did the conductivity of the alkali chamber first drop and then rise?
During the experimental study, there may be an error originating from the feed pump or the sensor. Experimental data are presented in their raw form.
(4) Compared with the second trial, the current drop point of the third trial was advanced. What is the reason?
It is thought to be due to the reasons given in Answer 3.
(5) The entire system had run for a total of six trials. When analyzing, the authors must focus on highlighting the phenomenon that is different from the previous trial, and at the same time explain in detail the reason for the phenomenon, rather than repeating the same data.
The aim of the study is to produce more concentrated acid-base by continuously incorporating the concentrate in the reverse osmosis system into the NF + EDBM system. Work continued on the same phenomena so that the evaluation data could be standardized. Studies were carried out at different times over 6 cycles and acid, base, pH, conductivity, pressure phenomena were evaluated.
(6) It is recommended that the authors provide the material balance calculation diagram of the entire process, including flow rate, conductivity, main product types and concentrations, etc.
These results are given in Table 3 of the manuscript.
(7) The authors need to conduct an economical analysis of the entire system, including operating costs, generating acid-base benefits, and so on. At the same time, a comparison should be made with the existing reverse osmosis concentrated brine treatment process to highlight the characteristics and advantages of the process.
This study focused on acid-base production as a target. Economic analysis is another study topic
Author Response
Thank you for your kind and polite comments on the manuscript, but previous comments by you and replies by us have been returned. In this case, the answers we gave earlier are the same.
It will be highly appreciated if you kindly provide us by your acceptance letter. We would like to thank for your cooperation in advance.
Please see the attachment,

Reviewer 2 Report
The authors made little effort to address the comments and bring the manuscript to an acceptable level for the scientific literature. Some questions were even answered in Turkish. The authors shold be given one more chance to address the concerns carefully.
Previous comment #2 should be revisited and errors/reproducibility demonstrated as required in any scientific literature.
Previous comment #4 should be revisited and the manuscript updated accordingly.
Previous comment #5 should be revisited and examples given beyond only EDBM.
Previous comment #6 should be revisited and the manuscript needs to be updated with some text that clarifies the tranferable aspects and gneral interest and use of the presented industrial research.
Previous comment #7 should be revisited and answers given in English and not Turkish.
Previous comment #8 should be revisited and examples given beyond only EDBM.
Previous comment #10 should be revisited. The authors misinterprested the comment. It has nothing to do with Table 1 and regulation values.
Author Response
Please see the attachment,
It will be highly appreciated if you kindly provide us by your acceptance letter. We would like to thank for your cooperation in advance.

Reviewer 3 Report
The authors have answered all the questions to my satisfaction. Comments and questions were properly addressed and the final manuscript clearly has been improved. It means that revised manuscript meets the criteria and in my opinion can be published as original paper in Membranes Journal.Author Response
Thank you for your kind and polite comments on the manuscript.
It will be highly appreciated if you kindly provide us by your acceptance letter. We would like to thank for your cooperation in advance.
Reviewer 4 Report
The manuscript was improved. The journal format will be realized in last step.
Author Response
Thank you for your kind and polite comments on the manuscript.
It will be highly appreciated if you kindly provide us by your acceptance letter. We would like to thank for your cooperation in advance.
Round 3
Reviewer 1 Report
It is recommended to accept.
Reviewer 2 Report
-